# Precision digital mapping of endogenous and induced genomic DNA breaks by INDUCE-seq

Felix M. Dobbs[1,4,5], Patrick van Eijk [1,5], Mick D. Fellows[2], Luisa Loiacono[2], Roberto Nitsch [3] & Simon H. Reed [1✉]

Understanding how breaks form and are repaired in the genome depends on the accurate measurement of the frequency and position of DNA double strand breaks (DSBs). This is crucial for identification of a chemical's DNA damage potential and for safe development of therapies, including genome editing technologies. Current DSB sequencing methods suffer from high background levels, the inability to accurately measure low frequency endogenous breaks and high sequencing costs. Here we describe INDUCE-seq, which overcomes these problems, detecting simultaneously the presence of low-level endogenous DSBs caused by physiological processes, and higher-level recurrent breaks induced by restriction enzymes or CRISPR-Cas nucleases. INDUCE-seq exploits an innovative NGS flow cell enrichment method, permitting the digital detection of breaks. It can therefore be used to determine the mechanism of DSB repair and to facilitate safe development of therapeutic genome editing. We further discuss how the method can be adapted to detect other genomic features.

[1] Division of Cancer and Genetics, School of Medicine, Cardiff University, Cardiff, UK. [2] Clinical Pharmacology and Safety Sciences, R&D, AstraZeneca, Cambridge, UK. [3] Clinical Pharmacology and Safety Sciences, R&D, AstraZeneca, Gothenburg, Sweden. [4] Present address: Unit AB303, Level 3, BioData Innovation Centre, Wellcome Genome Campus, Cambridge, Hinxton, UK. [5] These authors contributed equally: Felix M. Dobbs, Patrick van Eijk. ✉email: reedsh1@cardiff.ac.uk

DNA double-strand breaks (DSBs) are the most toxic of all DNA lesions, directly compromising genome stability. Other than causing cell death, failure to repair DSBs accurately can lead to a range of structural genomic alterations associated with carcinogenesis[1–3]. Low-level physiological breaks can arise sporadically due to normal cellular processes such as DNA replication, transcription, and alterations to chromatin structure[2]. DSBs are also induced at high frequencies as programmed events in specialised cell-types. Similarly, intentionally induced DSBs are also substrates of meiotic recombination that enables genetic diversification in the germline[4]. A variety of exogenous agents, such as ionising radiation, chemotherapeutic drugs and more recently, CRISPR genome editing technologies[5,6], are also potent inducers of DSBs.

Precise measurement of the full complement of genomic DSBs present in cells has been a major challenge to achieve at scale. This is because most physiological breaks are infrequent and occur stochastically throughout the genome. Existing sequencing-based methodologies typically use classic amplification-based sequencing libraries to measure DSBs. This causes distorted break measurements due to PCR amplification bias introduced during DSB-library preparation[7–10]. As a result, recurrent DSBs formed either endogenously, or by sequence-directed nucleases are represented more frequently than rare break sites[11]. These next generation sequencing (NGS) based methods fall broadly into three categories: indirect break labelling using proteins as a proxy for breaks (e.g. DISCOVER-seq, γH2AX ChIP-seq)[12,13], indirect labelling of repaired breaks (e.g. GUIDE-seq, HTGTS)[14,15] and finally, direct labelling of unrepaired break ends in situ (e.g. END-seq, BLESS, DSBCapture, BLISS, i-BLESS)[11,16–19]. All these methods employ PCR amplification during the standard DNA library preparation for sequencing[7–10]. To correct for the bias introduced by library amplification, others have employed different methods such as unique molecular identifier (UMI) correction or DSB spike-ins[20]. The method qDSB-seq uses a restriction enzyme induced DSB spike-in to try to normalise and quantify the breaks examined in the experimental samples[20]. This method relies on engineered cell-lines, or the introduction of a specific type of exogenous DSB spike-in. This is considered representative of the range of break types present in the cell sample, which permits the calculation of a correction factor that is subsequently used for normalisation of all breaks present in the sample. However, a linear relationship that can be observed for a single restriction enzyme induced spike-in is not representative of all types of DSBs which has previously been reported[20]. This limits the application of this technique to normalising the break numbers between different experiments but not within a sample. For most NGS applications, PCR amplification during library preparation does not present a problem. However, for accurate genome-wide measurement of specific features, such as DSBs, amplification introduces high levels of noise into the system, by misrepresenting the true frequency of breaks in the genome. To overcome the attenuation of the actual DSB signal by the noise associated with amplification, here we show the design of a DNA library preparation that eliminates PCR and permits enrichment of breaks directly on the Illumina flow cell. By eliminating the distortion associated with signal amplification during DSB measurement, we obtain a digital representation of genomic breaks, where one sequence read observed is derived from one labelled DSB-end originally present in the cell sample. This means that whenever multiple reads are detected at the same position, they are derived from different cells. For this reason, it is now possible to normalise and quantify DSBs based on cell number. Here, we describe the method INDUCE-seq for the direct, digital measurement of genomic DSBs. We demonstrate its ability to locate and characterise endogenous and induced DSBs caused by different nucleases including CRISPR-Cas9.

## Results

**INDUCE-seq: a method for the digital detection of DSBs.** Break measurement by INDUCE-seq is achieved via a two-stage, PCR-free library preparation (Fig. 1 and Supplementary Fig. 1a). Stage one consists of labelling in situ, end prepared DSBs via ligation of a full-length, chemically modified P5 adapter. In stage two,

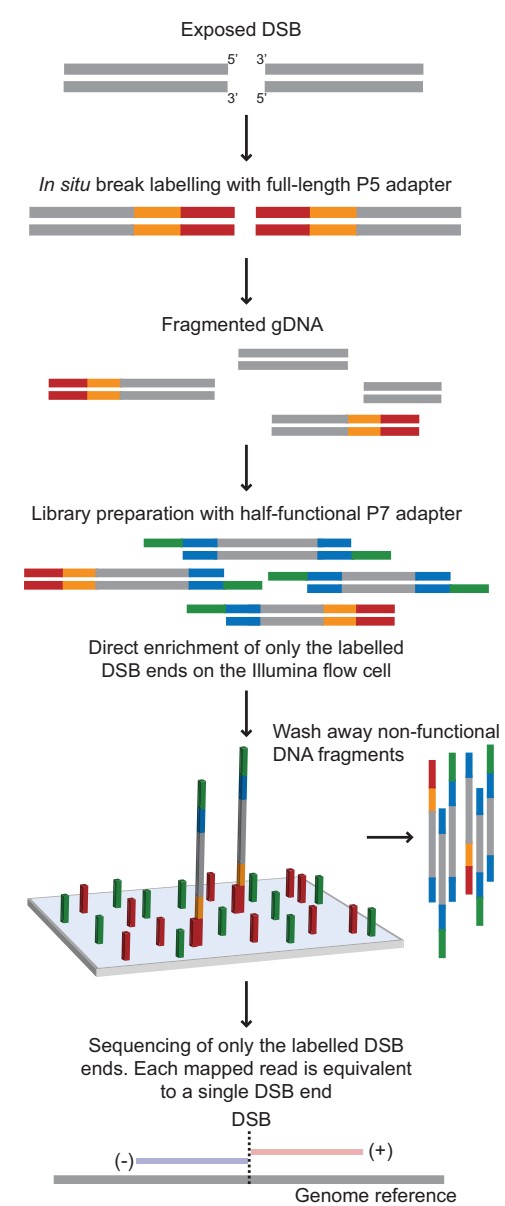

**Fig. 1 The INDUCE-seq workflow.** In situ break labelling in fixed and permeabilised cells is performed by ligating a full-length, chemically modified P5 sequencing adapter to end-prepared DSBs. Genomic DNA is then extracted, fragmented, end-prepared and ligated using a chemically modified half-functional P7 adapter. Resulting DNA libraries contain a mixture of functional DSB-labelled fragments (P5:P7) and non-functional genomic DNA fragments (P7:P7). Subsequent illumina sequencing of INDUCE-seq libraries enriches for DNA-labelled fragments and eliminates all other non-functional DNA. As the INDUCE-seq library preparation is PCR-free, each sequencing read obtained is equivalent to a single labelled DSB-end from a cell.

extracted, fragmented, and end-prepared genomic DNA is ligated using a second chemically modified, half-functional P7 adapter. The resulting DSB-labelled DNA fragments, which comprise both the P5 and half-functional P7 adapters can hybridise with the illumina flow cell and subsequently be sequenced. Remaining genomic DNA fragments that are not ligated to the P5 adapter are therefore not derived from a physiological break-end and are subsequently blocked by the ligation of two half-functional P7 adapters. This renders them non-functional, since they lack the sequence required to hybridise to the flow cell. This strategy enables the enrichment of functionally labelled, physiological DSB sequences on the flow cell and the elimination of all other genomic DNA fragments, preventing them from generating system noise. The avoidance of break-sequence amplification produces a sequencing output where a single sequencing read is equivalent to a single labelled DSB-end in the cell (compare Supplementary Fig. 1a, b, see Supplementary Fig. 2a). This important innovation generates a digital DNA break readout, enabling the direct detection and quantification of genomic DSBs by sequencing without the need for error-correction. Following in situ break labelling, currently available DSB detection methods BLISS, DSBCapture and END-seq, all employ an enrichment protocol to separate DSB-labelled DNA fragments from the excess, fragmented genomic DNA. This is followed by a PCR-based library preparation, and illumina sequencing (Supplementary Fig. 1b). This results in a readout where a single read is not equivalent to a single break. Therefore, PCR error-correction, such as UMI-correction, is employed to adjust for this[18] (Supplementary Fig. 1b and Supplementary Fig. 2b). Importantly, INDUCE-seq is compatible with any of the in situ DSB labelling protocols reported to date.

**In vitro detection of restriction enzyme-induced breaks**. To evaluate the performance of INDUCE-seq, we first examined the ability of the method to capture genome-wide DSBs induced by a high-fidelity HindIII restriction endonuclease in permeabilised, cross-linked HEK293T cells. This approach has been used previously to benchmark other DSB detection methods such as END-seq, BLISS and DSBCapture[11,17,18]. There are ~800,000 HindIII genomic positions that match the restriction sequence. However, each of these restriction sites will be cut with varying efficiencies, depending on their location in the genome and other factors. Therefore, the total number of breaks detected in the sample, can be used to calculate the average number of breaks per cell across the population of cells examined. We labelled DSBs in cross-linked HEK293T cells and prepared INDUCE-seq libraries to sequence labelled DSBs obtained from the genomic DNA extracted from ~25,000 cells, yielding ~148 M reads. This number is equivalent to the total number of breaks detected in the experimental sample. Therefore, INDUCE-seq detects an average of ~5900 breaks per cell (~148 M breaks divided by 25,000 cells). In untreated HEK293T cells, ~200 K breaks were detected. These represent background, endogenous breaks present in the cell sample. This equates to an average of ~2 endogenous breaks per cell (~200 K breaks divided by 100,000 cells). As shown in Fig. 2a, INDUCE-seq simultaneously detects highly recurrent HindIII-induced DSBs as well as lower-frequency endogenous DSBs (highlighted in red) within the same sample. Supplementary Fig. 2c also illustrates the simultaneous detection of recurrent and non-recurrent endogenous breaks in HEK293 cells. Furthermore, this data revealed that ~146 M of these reads map precisely at the expected cut site for ~782,000 HindIII restriction sites located within the genome. To determine the distribution of the reads obtained at HindIII sites detected, we ranked them according to the number of breaks observed from highest to lowest and plotted

the data in Fig. 2b. The top-ranked site detected by INDUCE-seq (i.e. the most frequently cut site in the cell population) has ~2500 reads observed. This means that cutting at this specific restriction site can be detected in ~10% of the 25,000 cells in the sample. At the other end of the scale, ~2000 HindIII sites were detected with just a single read, meaning that these sites are cut very infrequently, being detected in a single cell from the 25,000 cells sampled (see Fig. 2b). Plotting the cumulative break frequency illustrates the variation in HindIII site cutting efficiency by demonstrating that 50% of the total breaks detected are accounted for by 30% of the top-ranked HindIII sites. Collectively, these observations demonstrate a remarkable dynamic range for break detection achieved by INDUCE-seq across three orders of magnitude. This is made possible by the elimination of PCR amplification during sequencing library preparation, and the remarkable efficiency of sequencing flow-cell enrichment of labelled DSB breaks. These features permit the accurate, simultaneous detection and proportional measurement of both endogenous and induced DSBs within the same sample for the first time.

Direct end-labelling and data processing also allows the structure of breaks at base pair resolution to be examined. As with other end-capture methods, INDUCE-seq detects the precise HindIII cleavage pattern of two semi-overlapping symmetrical blocks of sequencing reads that map to the known HindIII cleavage positions on both strands of DNA (Fig. 2c). INDUCE-seq is unique because it precisely measures the frequency of reads on both sides of a break, and this can inform on the mechanism of break induction. Thus, INDUCE-seq can be used to precisely measure DSB-end structures at single-nucleotide resolution.

**Comparison of INDUCE-seq with an alternative DSB detection method**. To compare the performance of INDUCE-seq in capturing in situ, restriction enzyme- induced DSBs, we identified a previous study which performed a similar experiment using a different DSB detection method called DSBCapture. Whilst the experimental design was the same a different restriction enzyme, EcoRV, was used to induce DSBs in the DSBCapture study, whereas we used the HindIII enzyme[17]. Therefore, we used the relative frequency of restriction enzyme cutting to enable comparison. We found that INDUCE-seq identifies a similar proportion of all possible HindIII restriction sites (92.7%), compared to all possible EcoRV sites (93.7%) as identified by DSBCapture (Fig. 2d). Significantly, INDUCE-seq captures >90% of the HindIII restriction sites present in the genome despite using just ~25,000 cells, which is 800-fold fewer cells than used by DSBCapture. In addition, a greater proportion of INDUCE-seq reads were mapped to restriction sites (Fig. 2e). These data reveal that 96.7% of aligned reads were mapped to restriction sites, representing an improved efficiency over DSBCapture (81.1%). We also identified ~1 M DSBs at HindIII off-target sequences; sites that differ from the restriction sequence by one or two mismatching bases (Fig. 2f). The total number of HindIII-induced DSBs measured by INDUCE-seq ranged from ~146 M across ~780,000 on-target sites (see above), to just five DSBs at the lowest ranking off-target site (containing 2 mismatches) (Fig. 2f). INDUCE-seq therefore detects breaks across seven orders of magnitude, enhancing the dynamic range of break detection over current methods.

**Detecting DSBs in live cells using INDUCE-seq**. To further characterise INDUCE-seq, we measured induced DSBs in live cells by using the DIvA cell system[13]. We confirmed that INDUCE-seq is capable of accurately detecting breaks at AsiSI sites in DiVA cells (Fig. 3a). In this experiment, INDUCE-seq

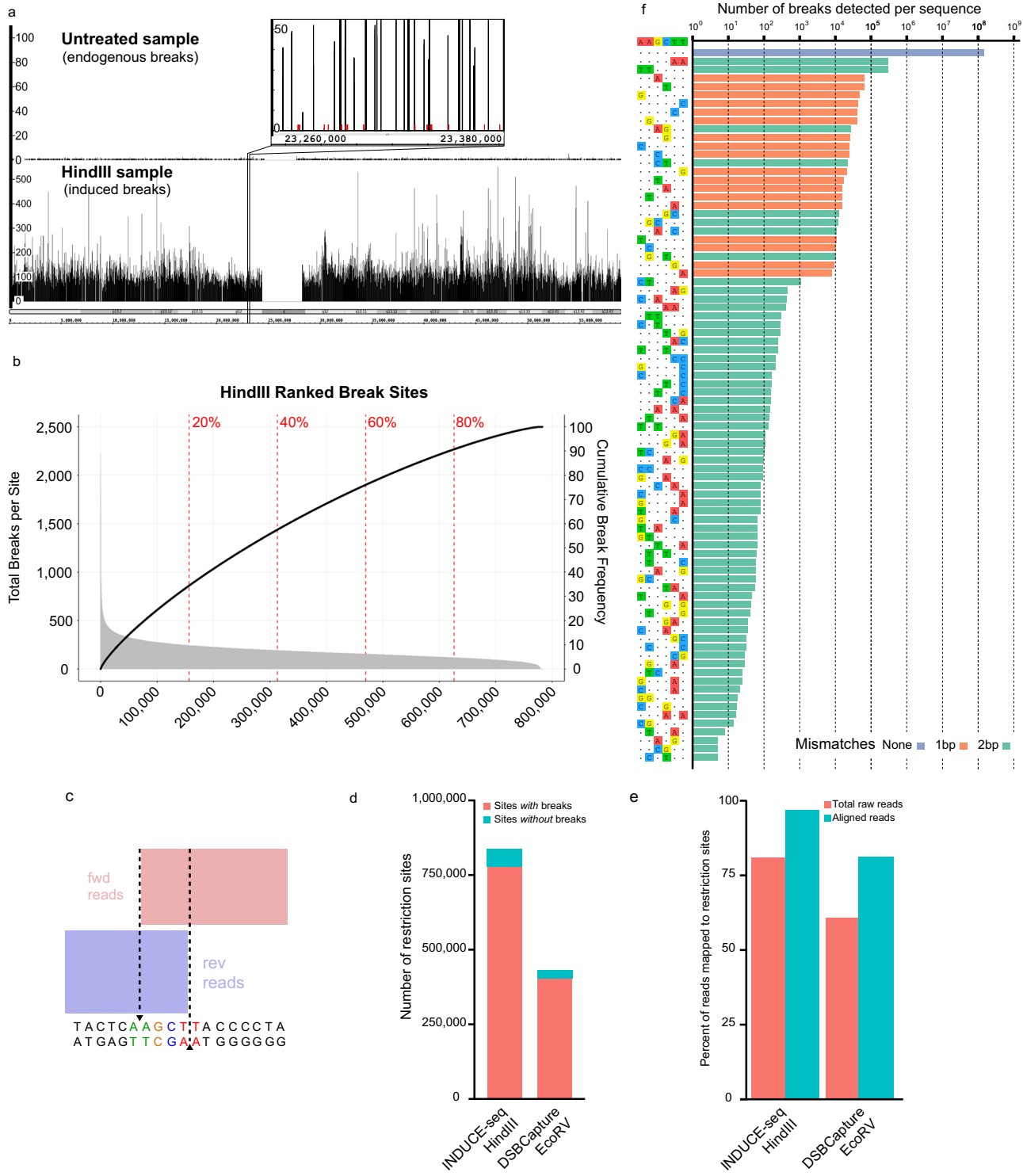

detected the presence of breaks at ~230 AsiSI sites, expanding the AsiSI sites discovered in live cells compared to the 214 sites reported by BLISS, and 121 sites reported by DSBCapture. It is important to note the difference between the raw sequencing output observed between these methods, which is enabled by the characteristics of INDUCE-seq (Fig. 3a). Figure 3a (bottom two panels) demonstrates the effect of PCR amplification on DSB sequencing output, which uncouples the relationship between breaks labelled in the cell and reads sequenced. This explains the need for some form of error-correction (e.g. UMI correction or

DSB spike-in) when using these methods. Importantly, this improvement in AsiSI site detection was achieved despite sequencing 40-fold fewer reads than a comparable DSBcapture experiment, and 23-fold fewer reads compared to a BLISS experiment[17,21]. This demonstrates that INDUCE-seq is significantly more sensitive, efficient, and cost effective than these other methods (Fig. 3b).

INDUCE-seq is designed to capture DNA double strand breaks in direct proportion to their frequency of occurrence within the starting cell population. Because each sequencing read obtained is

**Fig. 2 INDUCE-seq demonstrates improved sensitivity and dynamic range when compared to alternative DSB sequencing technologies. a** INDUCE-seq detects both highly recurrent induced DSBs and low-level endogenous DSBs simultaneously with high resolution. Genome browser view (IGB) of INDUCE-seq reads mapped to chromosome 19 and a 120 kb section from HEK293T cells following in situ cleavage with the restriction endonuclease HindIII. (**a**, top panel) untreated HEK293T DSB data on chromosome 19 reveals low level single endogenous breaks present in untreated cells. (Bottom panel) Highly recurrent enzyme-induced breaks represent the vast majority of reads when viewed at low resolution. An expanded panel shows a close-up region of 120 kb, demonstrating the simultaneous detection of HindIII induced breaks shown in black and endogenous breaks shown in red. **b** HindIII ranked break sites. Total number of breaks were plotted at each of the 782,602 HindIII restriction sites (Left, primary *y*-axis). The cumulative break frequency at HindIII sites was calculated and plotted as a percentage on the secondary *y*-axis. **c** Mapping of INDUCE-seq reads at a HindIII target site demonstrates the precision of single-nucleotide break mapping on both sides of the break. **d** and **e** Comparison between INDUCE-seq and DSBCapture in detecting in vitro cleaved restriction sites by the enzymes HindIII and EcoRV. **d** INDUCE-seq identifies a similar proportion of HindIII restriction sites (92.7%) to that identified by DSBCapture for EcoRV (93.7%). **e** A greater proportion of sequenced reads were mapped to restriction sites using INDUCE-seq compared to DSBCapture. **f** The dynamic range of induced DSB detection using INDUCE-seq. In addition to breaks identified at HindIII on-target sequences (AAGCTT), multiple 1 bp and 2 bp mismatching off-target sites were also identified. INDUCE-seq measured cumulative breaks spanning 8 orders of magnitude, from ~146 M breaks identified at HindIII on-target sites, down to the selected cut-off of 5 breaks identified per site at the least frequent off-targets. Source data are provided as a Source Data file.

derived from precisely one DSB present in a cell from the starting population, this means that the relationship between reads, and breaks is unity. It follows, therefore, that multiple reads located at the same genomic location (recurrent breaks), are typically derived from different cells in the starting population. Consequently, recurrent breaks identified by INDUCE-seq are a precise measure of the propensity of that genomic location to form breaks in the cell population being examined. These characteristic features of the data derived from INDUCE-seq, make it possible to quantify the breaks detected in relation to cell number in the starting sample. To illustrate this, firstly, we analysed INDUCE-seq data derived from DIvA cells and observed that the number of DSBs detected at AsiSI sites is directly proportional to the total number of breaks measured in the cell sample (Supplementary Fig. 3a). We found that this relationship is true whether using in silico subsets of the data (red data points) or using experimental repeats with different numbers of cells in the starting population (green data points). Importantly, the output is reproducible when comparing break numbers at AsiSI sites across four independent biological repeats (Supplementary Fig. 3b, c). However, examining the number of breaks observed at individually ranked AsiSI sites located at different genomic positions, reveals a dynamic range of 2 orders of magnitude, from one break per site, to hundreds of breaks per site, depending on the genomic location of the AsiSI site, as shown in Fig. 3c (bar chart, primary *y*-axis). Next, we ranked individual AsiSI sites by the number of breaks detected at them and calculated the cumulative break frequency expressed as a percentage across the rankings (see Fig. 3c, scatter secondary *y*-axis). This analysis revealed that 60% of the top-ranked AsiSI sites account for >90% of the total number of breaks at AsiSI sites as detected by INDUCE-seq. Plotting the data in this way revealed a striking asymptotic relationship (Fig. 3c). This clearly demonstrates the difference in enzyme cutting efficiency at different AsiSI sites depending on their position in the genome. To demonstrate the sensitivity of INDUCE-seq, we show that it is possible to detect 38 AsiSI sites that are cut with very low efficiency being detected only once in 400,000 cells sampled. This clearly demonstrates the diminishing returns of sequencing increased numbers of breaks on the detection of additional AsiSI sites in the genome (Fig. 3c). This further confirms that not all AsiSI sites are cut with equal efficiency. Others have also observed a similar phenomenon[20]. Expressing cumulative break frequency as a percentage in relation to break site ranking, makes it possible to estimate how likely a break event will occur at a specific genomic location, and how many cells would need to be sampled to detect such an event. We note the presence in the genome of ~1200 AsiSI restriction sites based on DNA sequence. However, our data shows the detection of 232 sites when sampled from a

total of 400,000 cells. Furthermore, cumulative break frequency plotting predicts that few additional AsiSI sites will be discovered by increasing the starting cell number. It is known that the AsiSI restriction enzyme is CpG methylation sensitive, meaning that sites located in CpG methylation islands are refractory to enzyme cutting. These methylation islands are located in dense chromatin regions that are not cut by DNaseI. Therefore, we plotted DNaseI hypersensitivity at AsiSI sites discovered by INDUCE-seq (*n* = 232) and then compared them to the DNaseI hypersensitivity at the sites which were not detected (*n* = 990). We observed DNaseI hypersensitivity uniquely at the cut AsiSI sites, whereas uncut AsiSI sites exhibit no DNaseI hypersensitivity (Supplementary Fig. 3d). These results demonstrate that AsiSI sites not detected by INDUCE-seq are located in closed chromatin, which is associated with methylated CpG islands, explaining why they are refractory to cutting by AsiSI and therefore are not detected by INDUCE-seq.

**Detection of CRISPR-Cas9 genome-editing induced breaks.** Having established the characteristics of break detection by INDUCE-seq, next we applied it to the detection of CRISPR-Cas9-induced on- and off-target DSBs in the genome. This analysis is of central importance in safety profiling for the development of CRISPR-based cell and gene therapies. Following RNP nucleofection of HEK293 cells with the extensively characterised *EMX1* sgRNA, DSBs were measured at 0, 7, 12, 24 and 30 h post-nucleofection in two independent biological replicates. Cas9-induced breaks were then analysed and as described above, ranking these nuclease-induced breaks in order of their recurrence in the sample, enables the unbiased identification of both on- and off-target sites simply by virtue of their frequency. This unique feature of INDUCE-seq makes this possible (Fig. 4a, on-target highlighted in red). Importantly, by comparing ranked break lists for the CRISPR-edited versus the untreated control sample, it is possible to identify endogenous (background) recurrent breaks in the genome that are not caused by *EMX1* editing. These common recurrent sites found in both samples are highlighted in blue in Fig. 4a. These represent fragile sites in the genome that occur endogenously in the cells examined. It is established that off-target editing induced by CRISPR-Cas9 is in part due to sequence degeneracy of the target site. However, it is also known that sequence alone is not sufficient predict off-target editing[22]. We therefore developed a sequence-based, off-target discovery pipeline, which is described in Supplementary Fig. 4. This enables us to apply a selection criterion to the sequence-based off-target predictions that takes full advantage of the empirical measurement of DSBs detected by INDUCE-seq. This is achieved by assessing the guide RNA sequence mismatches and

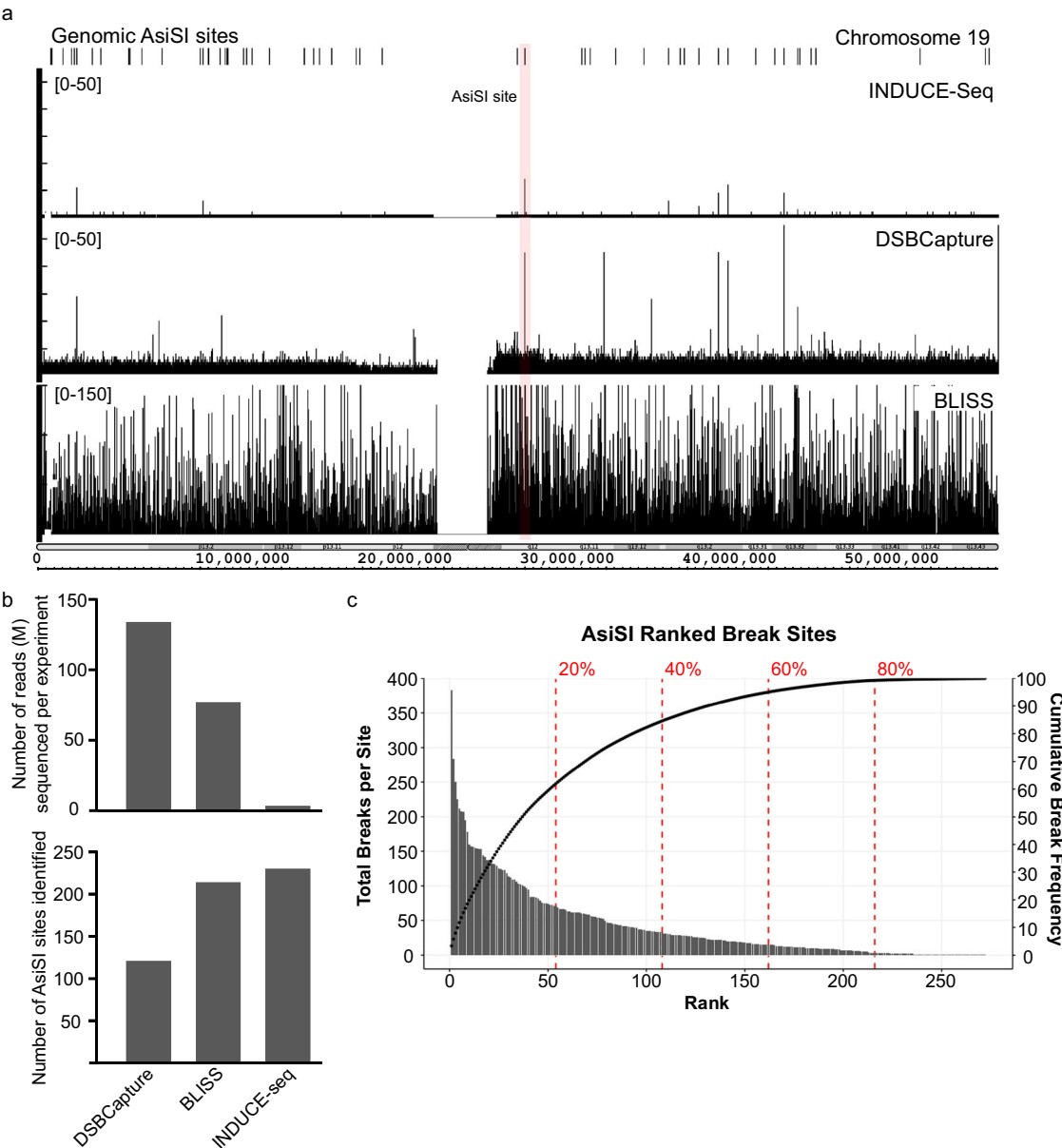

**Fig. 3 INDUCE-seq reveals the characteristics of AsiSI site cleavage in live DIvA cells. a** Genome browser view (IGB) comparing raw sequencing output of three different DSB capture methods used to detect induced breaks in live DIvA cells. The top panel indicates AsiSI recognition sequences present on chromosome 19. The second panel depicts the raw INDUCE-seq readout. The bottom two panels show the raw output of DSBCapture and BLISS.
**b** Comparison between INDUCE-seq, DSBCapture and BLISS in detecting AsiSI-induced breaks in live DIvA cells. The number of reads sequenced (**b**, top panel) is compared to the number of AsiSI sites identified in each experiment (**b**, bottom panel). **c** AsiSI sites detected by INDUCE-seq ranked by break number. Total number of breaks were plotted at each of the 278 AsiSI restriction sites (primary y-axis). The cumulative break frequency at AsiSI sites was calculated and plotted as a percentage on the secondary y-axis. Source data are provided as a Source Data file.

break occurrence at potential off-targets to create 32 distinct and partially overlapping filter conditions of varying stringency (Supplementary Fig. 4). It is now possible to apply these filter conditions to discover off-targets in treated and control samples. Supplementary Fig. 5 demonstrates the detection of off-targets across the varying filter conditions used and the degree of false positive detection is revealed by the non-treated control samples when viewed as a function of filter stringency. Supplementary Fig. 6a demonstrates the false positive discovery of off-target detection in relation to these different filter condition by plotting the reproducibility of off-target discovery (False Discovery Rate, FDR) as a function of total off-targets detected. Here, true positive off-targets were defined as the sites identified across the treated

sample group (Supplementary Fig. 5, pink bars), and false positive off-targets were defined as those identified across the control sample group (Supplementary Fig. 5, blue bars). This analysis can be applied to select the appropriate filter condition for off-target discovery for any given sgRNA by selecting the optimal combination of yield and FDR to maximise discovery power (bottom right quadrant of Supplementary Fig. 6a). For *EMX1*, filter condition 7 meet these criteria resulting in the highest number of off-targets discovered with the fewest number of false positives (Supplementary Fig. 6a, highlighted red) and was therefore selected for subsequent analysis (Fig. 4b). This figure shows the break number discovered at the on-target and each of the 60 off-targets detected (Fig. 4b, left panel). Detecting more off-targets

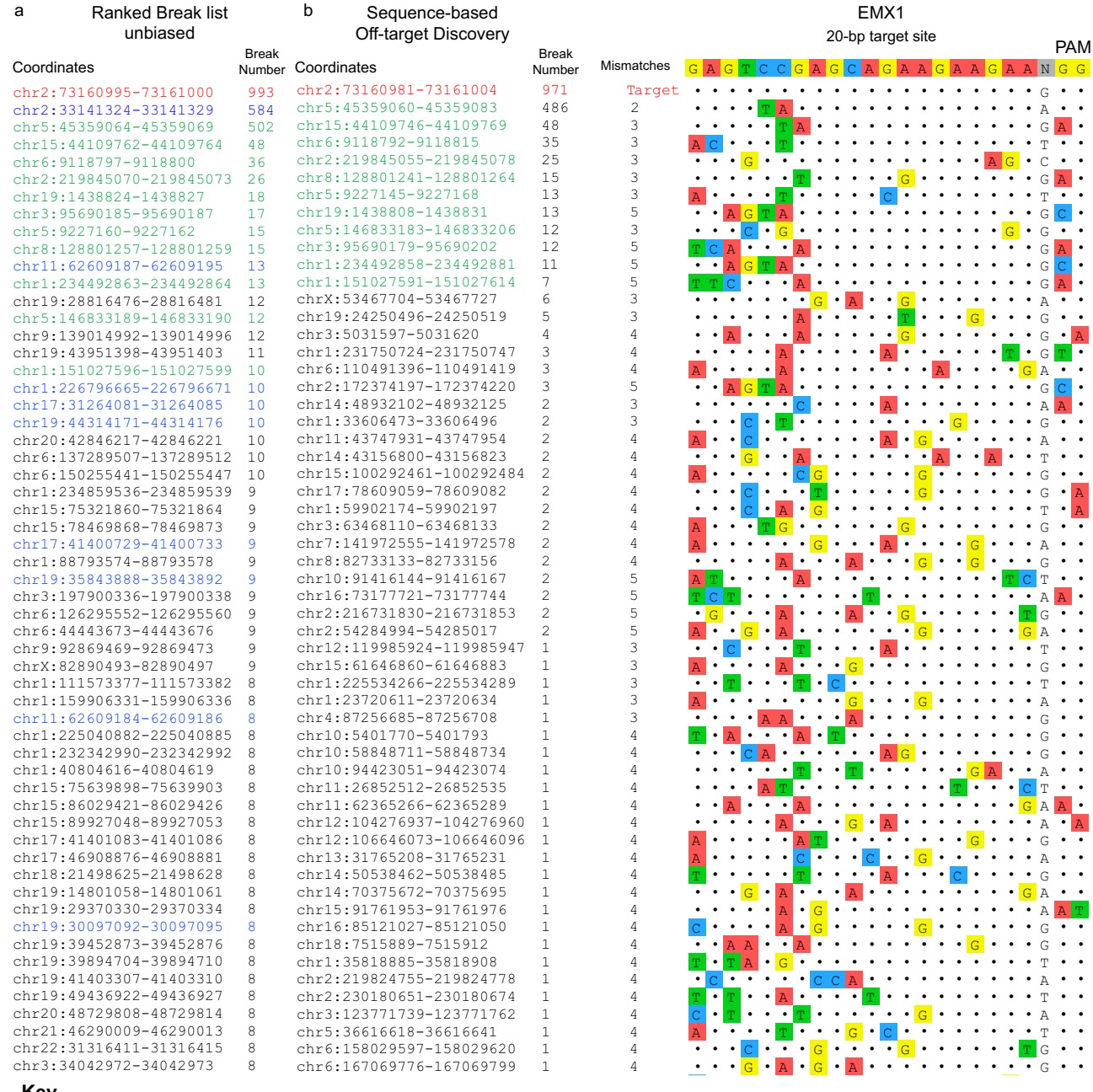

**Fig. 4 INDUCE-seq sensitively discovers and quantifies CRISPR-Cas9 induced on- and off-target DSBs in the genome. a** A ranked list of recurrent DSBs from the *EMX1* gene-edited data. The on-target is highlighted in red, while the recurrent breaks common to both treated and untreated control sample are highlighted in blue. The recurrent breaks common to both the rank-based list and the sequence-based off-target discovery list are highlighted in green. **b** Sequence-based off-target discovery list. (**b**, left panel) On- and multiple off-target sequences and the number of breaks identified using INDUCE-seq for the *EMX1* sgRNA. (**b**, right panel) Mismatching bases from the target sequence are highlighted and colour coded. Source data are provided as a Source Data file.

results in a higher rate of false discovery and vice versa. This explains the selection of filter condition 7 as being optimal for the discovery of editing induced breaks by *EMX1* during genome editing in this example. Optimal filter selection will be guide- and cell-type dependent. Figure 4b, right panel presents the off-target

mismatch plot, highlighting mismatches from the *EMX1* target sequence shown in colour.

Next, we compared the kinetics of break induction during *EMX1* gene editing across 5 time points. Figure 5a and Supplementary Fig. 6b and d show that the majority of both

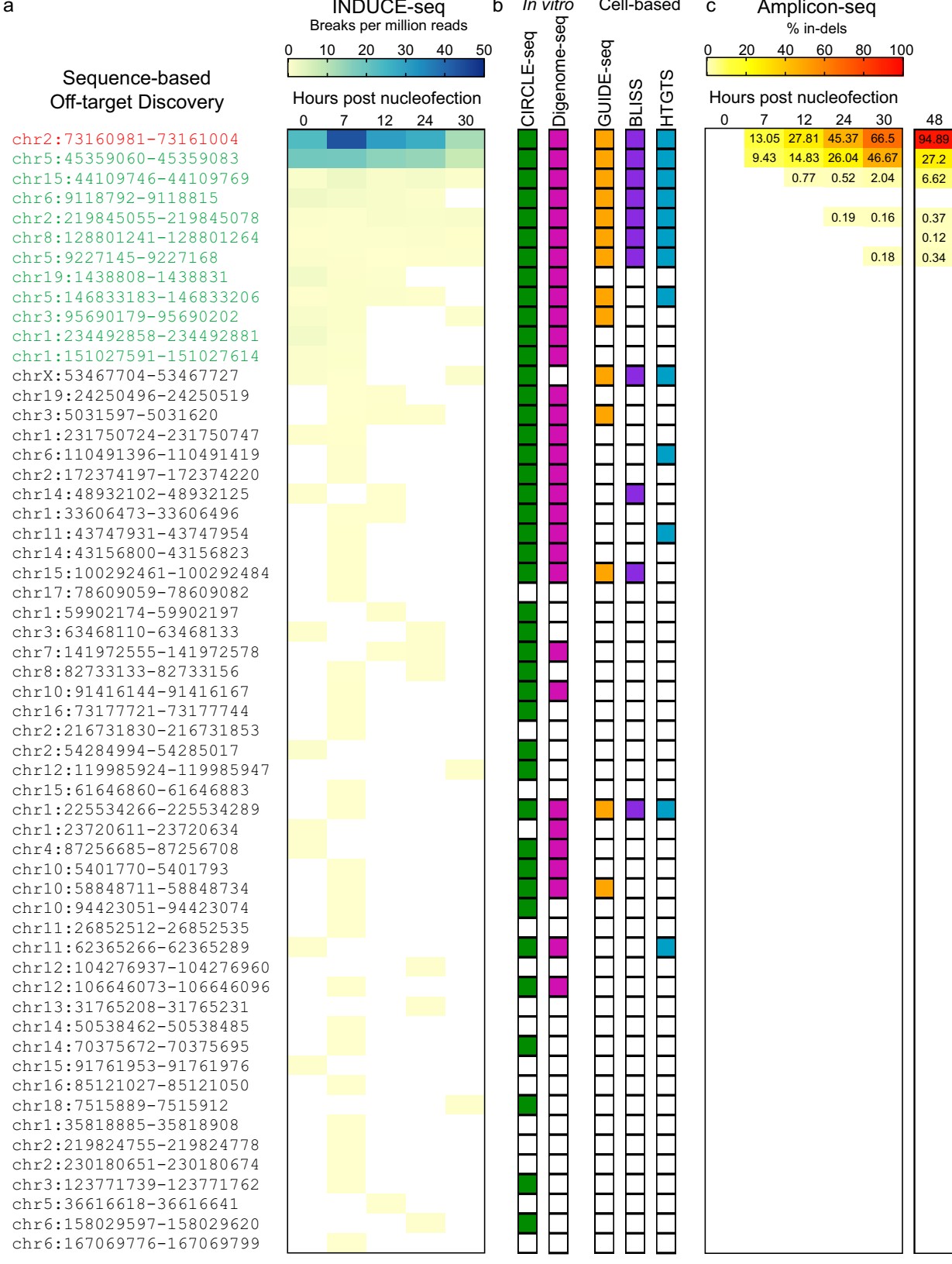

on- and off-target activity is observed immediately following nucleofection and during the early stages over the time course reported. Supplementary Fig. 6c shows the reproducibility of on- and off-target editing observed in two independent experiments. When compared to existing technologies, we find that INDUCE-

seq outperforms alternative cell-based methods GUIDE-seq, BLISS and HTGTS, as well as capturing several sites that were previously only identified using in vitro off-target discovery methodologies CIRCLE-seq and Digenome-seq (Fig. 5b). Importantly, INDUCE-seq also detects off-target break sites not

**Fig. 5 Kinetics of DSB formation and mutation induction following *EMX1* genome editing. a** INDUCE-seq reveals the kinetics of *EMX1*-induced DSB formation in a cell population during the editing process. Quantification of the number of nuclease-induced breaks detected per million reads for each sample revealed high Cas9 activity at both on- and off-targets immediately following cell nucleofection. **b** The comparison between off-targets identified by INDUCE-seq with established in vitro methods CIRCLE-seq and Digenome-seq, as well as cell-based methods GUIDE-seq, BLISS, and HTGTS. INDUCE-seq detects many off-targets that were previously only identifiable by in vitro methods. Substantially more off-target sites were identified than by any of the current cell-based methods. INDUCE-seq also identifies multiple off-targets not detected by any other method. **c** Amplicon-sequencing to measure the indel frequency at INDUCE-seq-identified off-targets. Four of the 60 off-targets discovered using INDUCE-seq were mutated with an indel frequency above the background false-discovery rate of 0.1% for amplicon-seq. (**c**, far-right) Indel frequency reported previously for *EMX1* 48 h post RNP nucleofection. Source data are provided as a Source Data file.

previously detected by any other method (Fig. 5b and Supplementary Fig. 7).

**Measuring mutations at INDUCE-seq detected off-target sites.** Finally, using the DNA from the samples analysed above, we measured the editing outcome at on- and off-target sites detected by INDUCE-seq, using amplicon sequencing[23]. This method accurately detects mutations with a sensitivity of only 0.1%[24–26]. This means that the mutational effect of editing can only be observed at sites that reach this threshold. Consequently, evidence of editing is only detected at the on-target, and four of the 60 off-target sites that were discovered using INDUCE-seq (Fig. 5c). This observation is in agreement with a previous study using GUIDE-seq for off-target detection[23], which identified five off-targets with indel frequencies >0.1% at 48 h post-nucleofection of HEK293 cells with *EMX1* RNP (Fig. 5c, 48 h). These results demonstrate that INDUCE-seq discovers CRISPR-induced DSBs at off-targets with a much higher sensitivity than indel detection using amplicon sequencing. This observation highlights the need for more sensitive methods for the detection of mutations, to accurately evaluate the safety of genome editing. We note that the sgRNA-specific cleavage pattern observed reflects the editing outcome at the on-target (Supplementary Fig. 8a) and top two off-target sites (Supplementary Fig. 8b and c). This raises the intriguing possibility of using the CRISPR-induced DSB pattern to model and predict the editing outcome.

## Discussion

We developed a PCR-free methodology to prepare DNA libraries for next generation sequencing of genomic DSBs. This advance, which exploits enrichment of break sequences using the Illumina flow cell, generates a digital output for measurement of breaks in cells. Our approach overcomes the problem of poor signal-to-noise ratios for DSB detection caused by PCR-amplification employed in standard NGS library preparation for the detection of genome-wide DSBs. Our innovative INDUCE-seq adapter design permits the sequencing flow cell to be used to enrich for labelled DSB sequences, thereby avoiding the need for their amplification by PCR. The resulting improved signal-to-noise ratio is instead achieved by eliminating the break ends generated during DNA fragmentation that are not associated with physiological DSBs found in cells. We demonstrate the characteristics of INDUCE-seq for measuring genomic DSBs in a range of different applications. We reveal its capacity to detect low-level endogenous, as well as high-level restriction enzyme-induced breaks simultaneously within the same sample. This has not been possible previously without the need for complex error-correction methods that each have their limitations. We compare INDUCE-seq with other currently available break detection methods and demonstrate the improved performance, scalability, ease of use, and cost effectiveness. These are all essential features of an assay that can be used for safety profiling of synthetic guides used in CRISPR genome editing, particularly as the field moves towards

the discovery and use of future gene editing systems. We demonstrate how INDUCE-seq compares to several of the current CRISPR off-target detection methods for the measurement of editing by the commonly used *EMX1* sgRNA. We reveal that INDUCE-seq identifies a significant number of off-target sites in addition to those reported previously by five other methods. INDUCE-seq may be a valuable method for safety profiling and synthetic guide RNA design for the future development of genome editing as a therapeutic modality. Finally, we note that this methodology can be adapted for the detection of a range of other genomic features that can be end-labelled in this way. Such features include genome-wide mutations, single strand breaks and gaps, as well as other types of DNA damage that can be converted into breaks and subsequently ligated using this combination of sequencing adapters. The further development of INDUCE-seq and its derivative assays could have significant implications in a range of different biomedical applications.

## Methods

**Cell culture and treatment.** HEK293, HEK293T (ATCC), and U2OS DIvA (a kind gift from the Legube lab) cells were cultured in DMEM (Life Technologies) supplemented with 10% FBS (Life Technologies) at 37 °C at 5% CO$_2$. HEK293 cells were nucleofected with 224 pmol RNP per $3.5 \times 10^5$ cells using a Lonza 4D-Nucleofector X unit with pulse code CM-130. Cells were harvested at 0, 7, 12, 24, and 30 h post nucleofection for INDUCE-seq processing. To stimulate AsiSI-dependent DSB induction, DIvA cells were treated with 300 nM 4OHT (Sigma, H7904) for 4 h. U2OS-DIvA cells were a gift from the Legube lab CBI, Toulouse, France. HEK293 (CRL-1573) and HEK293-T (CRL-3216) cells were obtained from ATCC.

**Cas9 protein and sgRNA.** The guide RNA targeting *EMX1* (GAGTCCGAGCAGAAGAAGAA) was synthesized as a full-length non-modified sgRNA oligonucleotide (Synthego). Cas9 protein was produced in-house (AstraZeneca) and contained an N-terminal 6xHN tag.

**INDUCE-seq method.** Cells were seeded to 96 well plates pre-coated with Poly-D-lysine (Greiner bio-one, 655940) at a density of ~$1 \times 10^5$/well and crosslinked in 4% PFA (Pierce, 28908) for 10 min at rt. Cells were washed in 1x PBS to remove formaldehyde and stored at 4 °C for up to 30 days. The INDUCE-seq method was initiated by permeabilising cells. Between incubation steps, cells were washed in 1x PBS at rt. Cells were permeabilised by incubation in Lysis buffer 1 (10 mM Tris-HCL pH 8, 10 mM NaCl, 1 mM EDTA, 0.2% Triton X-100, pH 8 at 4 °C) for one hour at 4 °C, followed by incubation in Lysis buffer 2 (10 mM Tris-HCL, 150 mM NaCl, 1 mM EDTA, 0.3% SDS, pH 8 at 25 °C) for one hour at 25 °C. Permeabilised cells were washed three times in 1x CutSmart® Buffer (NEB, B7204S) and blunt-end repaired using NEB Quick Blunting Kit (E1201L) + 100 µg/mL BSA in a final volume of 50 µL at rt for one hour. Cells were then washed three times in 1x CutSmart® Buffer and A-tailed using NEBNext® dA-Tailing Module (NEB, E6053L) in a final volume of 50 µL at 37 °C for 30 mins. A-tailed cells were washed three times in 1x CutSmart® buffer then incubated in 1x T4 DNA Ligase Buffer (NEB, B0202S) for 5 mins at rt. A-tailed ends were labelled by ligation using T4 DNA ligase (NEB, M0202M) + 0.4 µM Modified P5 adapter in a final volume of 50 µL at 16 °C for 16–20 h. Following ligation, excess P5 adapter was removed by washing cells 10 times in wash buffer at rt (10 mM Tris-HCL, 2 M NaCl, 2 mM EDTA, 0.5% Triton X-100, pH 8 at 25 °C), incubating for 2 mins each wash step. Cells were washed once in PBS and then once in nuclease free H$_2$O (IDT, 11-05-01-04). Genomic DNA was extracted by incubating cells in DNA extraction buffer (10 mM Tris-HCL, 100 mM NaCl, 50 mM EDTA, 1.0% SDS, pH 8 at 25 °C) + 1 mg/mL Proteinase K (Invitrogen, AM2584) in a final volume of 100 µL for 5 mins at 37 °C. The cell lysates were transferred to 1.5 mL Eppendorf RNA/DNA LoBind tubes (Fisher Scientific, 13-698-792) and incubated at 65 °C for 1 h,

shaking at 800 rpm. DNA was purified using Genomic DNA Clean & Concentrator™−10 (Zymo Research, D4010), and eluted using 100 μL Elution Buffer. DNA yield was assessed using 1 μL sample and Qubit DNA HS Kit (Invitrogen, Q32854) before proceeding to library preparation. Genomic DNA was fragmented to 300–500 bp using a Bioruptor Sonicator, and size selected using SPRI beads (GC Biotech, CNGS-0005) to remove fragments <150 bp. Fragmented and size-selected DNA was end-repaired using NEBNext® Ultra™ II DNA Module (NEB, E7546L). Fragmented and end-repaired DNA was added directly to the ligation reaction using NEBNext® Ultra™ II Ligation Module (NEB, E7595L) according to the manufacturer's instructions using 7.5 μM Modified half-functional P7 adapter and omitting USER enzyme addition. The ligated sequencing libraries were purified using SPRI beads. Libraries were purified twice more using SPRI beads, and size selected to remove fragments <200 bp to remove residual adapter DNA. Final clean libraries were quantified by qPCR using the KAPA Library Quantification Kit for Illumina® Platforms (Roche, 07960255001). Samples were pooled and concentrated to the desired volume for sequencing using a SpeedVac. Sequencing was performed on an Illumina NextSeq 500 using 1 x 75 bp High Output flow cell.

**INDUCE-seq adapters**. All modified INDUCE-seq adapter oligonucleotides were purchased from IDT. Single stranded oligonucleotides were annealed at a final concentration of 10 μM in Nuclease-free Duplex Buffer (IDT, 11-01-03-01) by heating to 95 °C for 5 minutes and slowly cooling to 25 °C using a thermocycler. INDUCE-seq P5 adapter: 5′-A*ATGATACGGCGACCACCGAGATCTA-CACTCTTTCCCTACACGACGCTCTTCCGATC*T-3′ and 5′-Phos-GATCG-GAAGAGCGTCGTGTAGGGAAAGAGTGTAGATCT CGGTGGTCGCCGTAT-CATT-spacerC3-3′. INDUCE-seq half-functional P7 adapter: 5′-Phos-GATCGGAAGAGCACACGTCTGAACTCCAGTCAC-spacerC3-3′ and 5′-C*AAGCAGAAGACGGCATACGAGAT[INDEX]GTGACTGGAGTTCA-GACGTGTGCTCTTCCGATC*T-3′ (*phosphorothioate bond). The index used here is 6 bp long.

**In situ DSB induction with HindIII**. Pilot INDUCE-seq experiments were performed by inducing DSBs in situ in HEK293T cells using the restriction enzyme HindIII-HF® (NEB, R3104S). This process was the same as described for the full INDUCE-seq method, with the addition of DSB induction prior to end blunting. Following cell permeabilization DSBs were induced using 50U HindIII-HF® in 1x CutSmart® Buffer in a final volume of 50 μL. Digestions were performed at 37 °C for 18 h.

**INDUCE-seq data analysis pipeline**. Demultiplexed FASTQ files were obtained and passed through Trim Galore! (http://www.bioinformatics.babraham.ac.uk/projects/trim_galore/) to remove the adapter sequence at the 3′ end of reads using the default settings. Following read alignment to the human reference genome (GRCh37/hg19) using BWA-mem[27], alignments mapped with a low alignment score (MAPQ < 30) were removed using SAMtools[28] and soft-clipped reads were filtered using a custom AWK script to ensure accurate DSB assignment. The resulting BAM files were converted into BED files using bam2bed function from bedtools[29], after which the list of read coordinates were filtered using regions of poor mappability, chromosome ends, and incomplete reference genome contigs, to remove these features from the data. DSB positions were assigned as the first 5′ nucleotide upstream of the read relative to strand orientation and were output as a breakends BED file. Care was taken to remove optical duplicates while retaining real recurrent DSB events. By maintaining each read ID, flow cell X and Y positional information was used to filter out optical duplicates using a custom AWK script. The final output was a BED file containing a list of quantified single nucleotide break positions.

**HindIII-induced DSB analysis in HEK293T cells**. The positions of HindIII target sites within hg19 were first predicted in silico using the tool SeqKit locate[30], allowing a maximum mismatch of 2 bp from the HindIII target sequence AAGCTT. The number of breaks overlapping with these predicted sites was calculated using bedtools intersect. To compare with the DSBCapture EcoRV experiment[17], the same coverage threshold of ≥5 breaks per site was used to define each HindIII induced break site.

**AsiSI-induced DSB detection and analysis in DIvA cells**. The positions of AsiSI target sites were calculated in the same way as for HindIII, however with no mismatches allowed and using the sequence GCGATCGC. As DIvA cells are female, sites present on the Y chromosome were removed leaving 1211 sites for chr1-X. To stringently calculate genuine AsiSI induced breaks, the 8 bp AsiSI site was reduced to 1 bp genomic intervals at the predicted break positions. This reduced each 8 bp genomic interval to two 1 bp intervals; at position 6 on the plus strand, and position 3 on the minus strand. Direct overlaps were then calculated between 1 bp breakend positions and the predicted AsiSI break sites using bedtools intersect. Matching strand orientation was required for each overlap to be considered a genuine AsiSI-induced break site.

**CRISPR off-target analysis pipeline**. Two sets of potential off-target sites for *EMX1* in hg19 were first predicted using the command line version of Cas-OFFinder[31], allowing up to 6 mismatches in the spacer and canonical PAM combined for the first set, and up to 7 mismatches for the second. Next, both sets of predicted sequences were filtered based on the mismatch number in the seed region, defined as the 12 nucleotides proximal to the PAM. Each set was filtered for up to 2, 3, 4 and 5 mismatches in the seed, generating a set of 8 files with different mismatch filtering parameters. To define CRISPR-induced DSBs, each 23 bp predicted site was first reduced to a 2 bp interval flanking the expected CRISPR break position, 3 bp upstream of the PAM. Overlaps were then calculated between these 2 bp expected break regions and the INDUCE-seq 1 bp breakend positions using bedtools intersect[29], returning a set of DBSs identified at expected CRISPR break sites. Finally, DSBs overlapping with CRISPR sites were filtered based on the site mismatch number and the number of breaks detected at the site. Sites possessing mismatches >n were required to have more than 1 DSB overlap to be retained as a genuine off-target site. Each set of break overlaps was filtered using a mismatch value of >2, >3, >4 and >5, resulting in a total of 32 filter conditions and off-target datasets for each INDUCE-seq sample.

**Calculating overlaps between CRISPR off-target detection methods**. *EMX1* off-target sites were compared with alternative methods GUIDE-seq, HTGTS, BLISS, CIRCLE-seq and Digenome-seq[14,15,18,32,33]. Genome interval files were generated for each respective off-target detection method. Overlaps of the *EMX1* off-targets detected by each method were calculated using bedtools intersect[29]. Genome interval files were generated for each respective off-target detection method. Overlaps of the *EMX1* off-targets detected by each method were calculated using bedtools intersect[29].

**Amplicon-seq validation of mutational outcome**. Amplicon sequencing DNA libraries were prepared using a custom panel of rhAmpSeq RNase-H dependent primers (IDT) that flank the INDUCE-seq identified off-targets for EMX1 (Supplementary Data 1). Multiplex PCR was carried out according to manufacturer's instructions using the rhAmpSeq HotStart Master Mix 1, the custom primer mix, and 10 ng of genomic DNA. PCR products were purified using SPRI beads and Illumina sequencing P5 and P7 index sequences were incorporated through a second multiplex PCR using rhAmpSeq HotStart Master Mix 2. Resulting sequencing libraries were pooled and sequenced using an Illumina NextSeq 550 Mid Output flow cell with 2 x 150 bp chemistry. Editing outcomes at the on- and off-targets were determined using CRISPResso software25 v2.0.32 with the following parameters: CRISPRessoPooled -q30 -ignore_substitutions–max_paired_end_reads_overlap 151. Indel frequencies were compared using CRISPRessoCompare.

**Reporting summary**. Further information on research design is available in the Nature Research Reporting Summary linked to this article.

## Data availability

All sequencing data related to this study have been deposited in the NCBI Sequence Read Archive at PRJNA636949 and are fully accessible: https://www.ncbi.nlm.nih.gov/bioproject/PRJNA636949/. Source data are provided with this paper.

## Code availability

All code used for secondary analysis is available on request. The INDUCE-seq code can be found at the following link: https://gitlab.com/brokenstringbio/induce-seq/induce-seq-manuscript/-/blob/main/03_induce-seq_v2.sh.

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

## Acknowledgements

We thank Jessica Downs and Hugang Feng, ICR, London, UK, for their insights and for providing U2OS DIvA cells. We thank Ross Chapman, Oxford University, Oxford, UK, for his insights and reading of the manuscript. We thank Steve Jackson, Cambridge University, Cambridge, UK, for constructive discussions. We acknowledge our colleagues at the Wales Gene Park for their insight, expertise, and technical support in generating the NGS data that assisted this research. Wales Gene Park is a Health and Care Research Wales funded infrastructure support group. F.M.D. is supported by a BBSRC/AstraZeneca CASE studentship (BB/P504841/1). P.V.E. is supported by the BBSRC (BB/R00756X/1).

## Author contributions

F.M.D. conceived and developed the INDUCE-seq method, conducted all experiments, performed bioinformatics analysis, interpreted results, and wrote the manuscript. P.V.E. contributed to the development of the INDUCE-seq method, contributed to the bioinformatics analysis, interpreted results, and wrote the manuscript. M.D.F. supervised the project and conceived of the study. L.L. contributed to conducting CRISPR experiments. R.N. supervised the CRISPR part of the project. S.H.R. supervised the project, conceived of the study, interpreted results, and wrote the manuscript.

## Competing interests

A UK patent application has been filed including work described in this publication. The authors declare the following competing interests: F.M.D., P.V.E. and S.H.R. are co-founders of Broken String Biosciences Ltd. M.D.F., L.L. and R.N. are employees and shareholders of AstraZeneca.
