## [Peer Review File · Nature Communications]

Reviewers' Comments:

Reviewer #1:

Remarks to the Author:

Dobbs et al. present a novel PCR-free method for detection and mapping of DNA double strand breaks. While the approach is surely innovative, there are some serious concerns that need to be addressed.

The authors claim that using INDUCE-seq, they can accurately measure the frequency of both endogenous and induced DSBs. However, both the description of the method and the data presented here suggest that DSB frequency is highly underestimated. The following points should be addressed before such claims can be made:

- As INDUCE-seq is a PCR-free method, any DNA fragments lost during library construction will lead to an underrepresentation of DSBs from the original cell population. This method includes a column-based DNA extraction step and multiple DNA purification steps using magnetic beads.

Significant loss of DNA is known to occur during both these procedures, and this should be addressed.

- For the validation of the INDUCE-seq method using HindIII digested DNA, the authors report detecting 'more than 3,000 enzyme-induced breaks per treated cell' (line 142). As pointed out later, breaks are detected at ~780,000 HindIII on-target sites. It seems unlikely that during 18 hours of digestion, the enzyme only cleaved a given restriction site in 1/260 cells. How do the authors explain this massive discrepancy between the numbers of expected and observed breaks?

- How was the average number of breaks/cell calculated and how many cells were used per sample? Neither is explained in the manuscript, and this information should be included.

Have the authors assessed the effect of varying the number of cells used per sample on quality of resulting data?

Do HindIII sites without breaks have something in common compared to sites with breaks? Is the lack of reads at those sites pure chance, or does it reflect actual lack of enzymatic cleavage at these sites?

What is the average break frequency at AsisI sites, as detected by INDUCE-seq? How does this compare to BLISS and DSBCapture?

How reproducible is INDUCE-seq? How well does the number of reads at individual HindIII/AsisI sites correlate between replicate experiments?

The authors refer to detection of endogenous DNA breaks, but do not discuss the possibility that at least a fraction of these breaks are induced experimentally, e.g. during formaldehyde fixation, and handling of cells/nuclei. This needs to be discussed in the manuscript and proof must be given that these are bona fide endogenous breaks as claimed.

For Figure 2e, please present the data as percentage of restriction sites with(out) breaks instead of absolute numbers. Total numbers can be indicated in either the figure or figure legend.

For Figures 2a and 2b, and Supplementary Figure 3, please indicate genomic coordinates of the windows shown.

In the top panel of Figure 2g, what exactly is displayed on the y-axis? It seems unlikely that this is the exact number of reads sequenced per library. Is it millions of reads?

Are the DSB hotspots shown in Supplementary Figure 3 common fragile sites? If so, this should be clearly stated. If not, do the authors detect enrichment of reads in known common fragile sites?

Supplementary Figure 5: why are off-target sites with seed mismatches grouped into $\leq x$ bp categories, instead of each category reflecting a distinct number of mismatches as to avoid overlap between categories?

Detection of Cas9 off-target cleavage should be discussed in greater depth in the text. How does the number of mismatches in off-target sites affect break level at these sites? What is the impact of the position of these mismatches related to the seed sequence? How does this compare to existing literature related to cleavage at off-target sites?

There are no references to Supplementary Figure 6b and 6c in the text.

What was the rationale for using filter condition 7 for subsequent analysis shown in Figure 3a?

Based on Supplementary Figure 7, more off-target sites were detected using BLISS than any other method shown here, including INDUCE-seq. Why then do the authors state that INDUCE-seq 'significantly outperforms' BLISS?

How many libraries were pooled for a single sequencing run? The authors claim that INDUCE-seq offers improved cost effectiveness, but show no direct comparisons of cost between different DSB mapping techniques.

Line 428: 'High Capacity flow cell' should presumably read 'High Output flow cell'?

Please indicate the length of the index present in the P7 adapter.

If DNA ends have a 3' overhang rather than 5' overhang, how is the fill in reaction going to work? Are the ends blunted in this case?

Can the authors explain better why they use the half-functional P7 adapter in their method?

Reviewer #2:

Remarks to the Author:

The Dobbs et al. manuscript proposes INDUCE-Seq, a novel method for mapping DSBs by sequencing. INDUCE-Seq does not include a typical enrichment and amplification step, instead relying on the cluster generation procedure, an integral step of Illumina sequence, to achieve also selection of reads originating from DSBs. This is an elegant idea, which allows for streamlining of DSB mapping via sequencing.

An important weakness of the paper though are the exaggerated claims of significance and novelty and the fact that some of the conclusions are unsupported by the data. For example, the authors incorrectly claim that precise DSB quantification (reported in Zhu et al., Nature Communications, 2019) or detection of low-frequency, endogenous DSBs (reported in Biernacka et al., Communications Biology 2018) were not previously possible. More generally, the authors both describe and visualize (Figure 1 bottom right) existing DSB mapping methods as noisy, which was true several years ago (e.g. original BLESS protocol, 2013), but is not true currently. Recent DSB mapping methods exhibit low noise (e.g. END-Seq allows to detect breaks after 1:10,000 dilution (Canela et al., 2016), i-BLESS after 1:100,000 (Biernacka et al., 2018)); they were also proven to be able to detect even very rare endogenous breaks, e.g. breaks induced by G-quadruplexes in vivo (i-BLESS). Precise DSB quantification is also now achievable through a straightforward approach, described in Nature Communications (Zhu et al., 2019).

Moreover, eliminating DSB enrichment and PCR amplification from current DSB mapping procedures, although definitely an improvement, is not as impactful as the authors suggest, since it is not true that the amplification is causing high-noise level in the newer methods (END-Seq, i-BLESS). Moreover, eliminating streptavidin DSB-enrichment and PCR amplification is not huge in terms of saved time or costs. Typical DSB mapping protocols are multiday (DSBcapture ~8 days, BLISS ~4 days, i-BLESS ~5 days), while streptavidin enrichment takes half an hour (3 hours with removal), PCR amplification - 2h, both are inexpensive.

The authors also repeatedly claim that their method is quantitative, that they accurately quantified breaks of various frequencies, etc – offering no validations whatsoever of these claims. It is not

even indicated how the numbers of DSBs per cell plotted in Fig. 2c were computed. Estimating DSBs per cells by dividing reads originating from unique breaks by the number of cells used, as proposed in BLISS (my guess how the numbers in Fig. 2c were calculated) may lead to very inaccurate results, as shown in Zhu et al, Nat. Comm. 2019, Fig. 2g.

The authors seem to believe that they achieved the perfect 100% efficiency of “1DSB=1 sequenced read”, but they did not attempt to verify it in any way. Detecting breaks during sequencing is a complex, multi-stage process with many challenges. First, DSB ends have to be labeled efficiently. It is not clear how efficiently INDUCE-Seq is doing it. For example, it has been shown that digestion with proteinase K before labeling (not used in INDUCE-Seq) improves the efficiency of labeling substantially (Biernacka et al., 2018, i-BLESS). Second, introduction of artefactual breaks has to be minimized. Initially, it was believed that fixation with formaldehyde helps to protect DNA integrity, but one of the innovations of the END-Seq (Canela et al., 2016) was to eliminate formaldehyde fixation, which allowed to significantly reduce noise. Usage of formaldehyde was again linked with higher noise in Biernacka et al., 2018. In INDUCE-Seq, fixation with 4% formaldehyde is used. Thirdly, not every DNA fragment will lead to a correct cluster on an Illumina flowcell. Sometimes more than one read will be present in a cluster (and such clusters are likely to be later rejected), sometimes one read will result in two neighboring clusters, etc. The authors allude to these kinds of problems only once and rather cryptically in the methods section (“optical duplicates were removed”), but they do not quantify how efficient really their labeling procedure is, nor do they mention it anywhere in discussion as a potential limitation.

Moreover, the comparison with DSBcapture shown in Fig. 2c and 2d is meaningless and even misleading. It is true that percentage of reads mapped to restriction site of an endonuclease when fully digested is used to compare efficiencies of DSB detecting methods, but it only makes sense when the same enzyme is used for both methods! As the authors state correctly, observed DSBs not mapping to restriction sites can include both endogenous DSBs and noise. Of course, depending on restriction enzyme, the same 10 endogenous DSBs will be a different percentage of a total signal, for restriction enzyme inducing likely 1,000s of DSBs and for an infrequent cutter. Here, since EcoRV has substantially fewer restriction sites it is unsurprising that percentage of reads mapping to them are a bit lower than for HindIII; the authors will likely get the same conclusion if they used both HindIII and EcoRV themselves. The figure 2c and d had to be removed as meaningless, or INDUCE-Seq has to be repeated with EcoRV, which is an easily commercially obtainable restriction enzyme.

Overall, removing PCR amplification from DSB mapping pipeline, as the authors propose, is a good idea, since Illumina technology allows it. It is also likely that INDUCE-Seq is highly sensitive, as shown by results of amplicon sequencing. However, it was not proven that it is superior to the current method, other than requiring less starting material. To be published, numerous inaccuracies have to be corrected, some specific examples are listed below.

Lines 76-78 “Existing methodologies typically measure DSBs reliably only when they exist at recurrent ‘hotspots’, or when they are induced at defined genomic-locations by sequence-directed nucleases.” Not true, for examples see i-BLESS (2018), proof of reproducibility in qDSB-Seq (2019).

Lines 87-88 “This well-known phenomena makes the quantification of genomic DSBs impossible”. This statement is based on papers published in 2011-2015 and directly contradicted by newer results (qDSB-Seq, 2019).

Bottom right of Figure 1 is misleading, the authors suggest that the current DSB mapping methods are so noisy that based on the read mapping, one cannot even confidently tell where the DSB was located, which is not true for well-defined DSBs tested with the newer methods (e.g. Fig 1d of i-BLESS).

Lines 122-125 “Therefore, PCR error-correction, such as a unique molecular identifier (UMI) is required to attempt DSB quantification”. DSB quantification can be simply achieved by employing spike-in DSB used to calibrate studied breaks, as validated in Zhu et al. Nat. Comm. 2019. That approach proved much more accurate than an earlier attempt to use UMIs (BLISS) (Zhu et al.

2019).

Reviewer #1 (Remarks to the Author):

Dobbs et al. present a novel PCR-free method for detection and mapping of DNA double strand breaks. While the approach is surely innovative, there are some serious concerns that need to be addressed.

The authors claim that using INDUCE-seq, they can accurately measure the frequency of both endogenous and induced DSBs. However, both the description of the method and the data presented here suggest that DSB frequency is highly underestimated.

This reviewer suggests that our explanation of the methodology, and the way we report the results, suggest that INDUCE-seq significantly underestimates the actual frequency of DSBs present in cells. We acknowledge that the way we initially communicated our report contributes to this misconception. Therefore, we have extensively revised and expanded key elements of the manuscript to rectify the problem.

Two factors contribute to the sense that INDUCE-seq underestimates DSB detection. The first relates to the unique, PCR-free INDUCE-seq library preparation used for labelling DSBs in cells *in situ*, and subsequently following genomic DNA extraction. This results in a fundamentally different sequencing output when compared to all other PCR-based NGS library preparations used for measuring DSBs in cells. The second reason for the sense of underestimation of breaks relates to a misunderstanding of the expected level of restriction enzyme-induced breaks when permeabilised, cross-linked cells are treated with a restriction enzyme. Cutting efficiency is much lower in this situation, which is very different from digesting naked DNA with a restriction enzyme *in vitro* where cutting efficiency is typically much higher. These important points and others are addressed in detail below, and have been rectified in the revised manuscript.

The following points should be addressed before such claims can be made:

- As INDUCE-seq is a PCR-free method, any DNA fragments lost during library construction will lead to an underrepresentation of DSBs from the original cell population. This method includes a column-based DNA extraction step and multiple DNA purification steps using magnetic beads. Significant loss of DNA is known to occur during both these procedures, and this should be addressed.

We acknowledge that the labelling of DSBs and the DNA extraction process during the construction of INDUCE-seq libraries cannot be 100% efficient, meaning that it is impossible to capture every break present in each of the cells in the population. However, this is true of any NGS-based DSB assay – in all cases, some losses are inevitable. We observed the expected levels of efficiency for each of the steps performed during INDUCE-seq break labelling, and the extraction of genomic DNA. Moreover, it should be acknowledged that PCR amplification can never compensate for break fragment losses caused by inefficient initial labelling of breaks in the cell population. In this regard it is noteworthy that INDUCE-seq is agnostic to the initial break labelling protocol used, meaning that any of the reported methods for preparing cells for the initial step of break labelling can be employed. We chose to adapt the BLISS *in situ* labelling protocol as one example. Finally, it is widely acknowledged that PCR amplification of labelled breaks introduces additional noise into the system, caused by biases in PCR efficiency of labelled-break sequences derived from different genomic locations. We have now included Figure 3a that demonstrates the extent of this noise by comparing the raw sequencing output from INDUCE-seq with two PCR-based methods.

We have revised our report so that the significance of eliminating PCR-induced biases from the reads obtained from labelled breaks can now be better understood. We've now demonstrated why INDUCE-seq represents an important advance in the precision of DSB measurements in cells. Briefly here, this can be explained as follows: existing PCR-based break detection methods provide an indirect, amplified representation of the original (true) breaks present in the cell. This leads to vastly elevated numbers of sequence reads obtained that are derived from the actual (true) break number present in cells. This distortion persists even after the use of various methods that attempt to correct for PCR amplification (eg UMI correction used by BLISS). PCR amplification therefore alters the relationship between the sequencing reads observed following break labelling, and the underlying (true) number of breaks originally present in cells. Because of this distortion, the relationship between reads obtained and the true break number is lost. To try and restore this lost relationship, the method qDSB-seq employs a DNA break 'spike-in' at specific, predetermined genomic loci. This provides a frame of reference based on restriction enzyme-induced break numbers at known genomic locations to calculate a correction factor (referred to as the proportionality co-efficient ' α '). Primarily, this correction factor permits normalisation of breaks between cell samples, but it is also applied to correct for, and estimate the number of 'true' breaks present per cell (However, see also our response to Reviewer 2 for further details regarding qDSB normalisation and the limitation of the use of this method for correcting for, and calculating numbers of breaks per cell).

In this report, we demonstrate that the novel design of the INDUCE-seq adapters permits the sequencing of individual labelled DSBs from cells, which hybridise with remarkable efficiency, and are therefore enriched on the sequencing flow cell. This key feature of the design eliminates the need for any PCR amplification during the library preparation. Consequently, this obviates the need for any complex error-correction methods for the following reason: every sequencing read reported by INDUCE-seq is derived from an individual DSB present in the genome of a single cell in the population of cells that was analysed. The corollary to this is that breaks detected at the same precise genomic location (recurrent breaks) are derived from different cells in the starting population. Therefore, any recurrent breaks reported by INDUCE-seq accurately measures and reflects the propensity of that genomic location to form breaks in the cells examined. We now communicate this

fundamental point more precisely in the manuscript, and this also enables us to address other related concerns raised by the reviewers (see detailed responses in the sections below).

- For the validation of the INDUCE-seq method using HindIII digested DNA, the authors report detecting 'more than 3,000 enzyme-induced breaks per treated cell' (line 142). As pointed out later, breaks are detected at ~780,000 HindIII on-target sites. It seems unlikely that during 18 hours of digestion, the enzyme only cleaved a given restriction site in 1/260 cells. How do the authors explain this massive discrepancy between the numbers of expected and observed breaks?

We have completely revised this section of the manuscript and conducted further experiments to better explain the HindIII results. The reviewer expects to observe highly efficient cutting at all possible HindIII restriction sites located throughout the entire genome in permeabilised cells. Perhaps such a high cutting efficiency might be predicted when digesting purified plasmid DNA *in vitro*. However, this is not the case when treating cross-linked, permeabilised cells *in situ*, as we now clearly explain in the revised manuscript. We show the variation in the frequency of restriction site cutting at the different HindIII sites throughout the genome (Figure 2B), and furthermore we demonstrate 'off-target' cutting at sites with 1 or 2 mismatches (Figure 2f). This demonstrates the extraordinary sensitivity of INDUCE-seq in detecting induced breaks, and this is now explained in detail in the revised manuscript. The reviewer notes their surprise at their correct assumption, and subsequent calculation that on average, HindIII enzyme cutting at a single given restriction site is observed in only 1 out of 260 cells. Intuitively, if considering this proposition based on efficiency of enzyme cutting, this observation appears to be too low. However, this is not the case if one considers this proposition as a sampling problem. In this case, the problem can be stated thus: on the average, if a single HindIII restriction site is selected from a total of ~780,000 possible HindIII restriction sites present in the genome, how many cells from a starting population of 25,000 need to be sampled to detect one break at that selected single site? The answer in this case is just 260 cells from the initial 25,000 cells sampled. This demonstrates the remarkable sensitivity of INDUCE-seq, and this nuance is now explained in detail in the revised manuscript.

- How was the average number of breaks/cell calculated and how many cells were used per sample? Neither is explained in the manuscript, and this information should be included.

We've now included a full explanation of how breaks/cell were calculated and included these in the manuscript. Briefly:

Starting cell number for HindIII treated cells: 25,000

Number of total reads/breaks: 148M

Number of reads at HindIII sites (no mismatches) in treated cells: 146M

Number of reads at HindIII sites with one or more mismatches: 1M

Number of HindIII sites with breaks detected: 782,602

Therefore:

Average number of breaks/site in HindIII-treated cells: 187

Average number of total breaks/cell: 5,900

Number of breaks in untreated cells: 0.2M

Number of untreated cells: 100K

Number of endogenous breaks/cell in untreated sample: 2

Have the authors assessed the effect of varying the number of cells used per sample on quality of resulting data?

Yes, we have tested this by adding increasing numbers of cells by pooling multiple samples. The same effect can be achieved by random sub-sampling from a single experiment. We've now included cumulative break frequency plots to demonstrate the effect of adding additional cells, and by sub-sampling within a single experiment. The data is presented in Supplementary Figure 3a and 3b).

Do HindIII sites without breaks have something in common compared to sites with breaks? Is the lack of reads at those sites pure chance, or does it reflect actual lack of enzymatic cleavage at these sites?

We have addressed this point in detail in the revised manuscript. Plotting total breaks detected per HindIII restriction site in rank order explains the variation in cutting efficiency observed at all sites throughout the genome. Some HindIII sites are cut much more efficiently than others (varying by over 3 orders of magnitude). By plotting the cumulative break frequency, the data shows the detection of around 2,000 unique HindIII sites present in the genome that are cut just once in the sample of 25,000 cells. The cumulative break frequency trend line suggests that sampling higher numbers of cells is likely to identify additional novel HindIII sites that are cut with correspondingly lower cutting efficiency. If increasing cell numbers were to be sampled, and this trend line were to reach a plateau prior to all possible HindIII restriction sequences being cut, this would indicate that certain HindIII sites present in the genome are refractory to cutting (compare this with observations made for AsiSI site cutting efficiency (shown in Figure 3C)). The evidence indicates that the lack of reads we observed at uncut HindIII sites is due to the extremely low cutting efficiency at these particular sites, and that increasing the initial cell sample size should reveal additional novel sites, albeit with diminishing returns for novel sites detected versus increased cells sampled.

What is the average break frequency at AsisI sites, as detected by INDUCE-seq? How does this compare to BLISS and DSBCapture?

This point is now fully addressed in the revised section relating to Figure 3 and expanded in Supplementary Figure 2a and 2b and Supplementary Figure 3.

How reproducible is INDUCE-seq? How well does the number of reads at individual HindIII/AsisI sites correlate between replicate experiments?

Reproducibility is now reported in full in the revised text and additional figures (eg Supplementary Figure 3A,B and C) to demonstrate INDUCE-seq's reproducibility, which is described in a variety of different ways.

The authors refer to detection of endogenous DNA breaks, but do not discuss the possibility that at least a fraction of these breaks are induced experimentally, e.g. during formaldehyde fixation, and handling of cells/nuclei. This needs to be discussed in the manuscript and proof must be given that these are bona fide endogenous breaks as claimed.

It is possible that a small number of the endogenous double strand breaks detected by INDUCE-seq could be due to non-physiological breaks produced during cell sample preparation. However, this is true of any of the methods for preparing cells for break labelling, and it is noteworthy that INDUCE-seq is agnostic to the preparation of cells prior to break labelling. However, assuming that non-physiological, experimentally-induced breaks are randomly distributed throughout the genome, then the reproducible nature of endogenous breaks observed in our samples indicates that physiological endogenous breaks predominate.

For Figure 2e, please present the data as percentage of restriction sites with(out) breaks instead of absolute numbers. Total numbers can be indicated in either the figure or figure legend.

We have revised this section accordingly to improve understanding.

For Figures 2a and 2b, and Supplementary Figure 3, please indicate genomic coordinates of the windows shown.

We have included this information as requested.

In the top panel of Figure 2g, what exactly is displayed on the y-axis? It seems unlikely that this is the exact number of reads sequenced per library. Is it millions of reads?

We apologise for this omission. It is indeed millions of reads. We have revised this section extensively to improve the key elements addressed.

Are the DSB hotspots shown in Supplementary Figure 3 common fragile sites? If so, this should be clearly stated. If not, do the authors detect enrichment of reads in known common fragile sites?

We have revised this section to better explain the features illustrated. These are not 'common fragile sites', which are not easy to define precisely as they are cell-type specific. We have identified recurrent breaks at certain previously reported common fragile sites in other cell types, but we do not report these in this manuscript.

Supplementary Figure 5: why are off-target sites with seed mismatches grouped into $\leq x$ bp categories, instead of each category reflecting a distinct number of mismatches as to avoid overlap between categories?

We have expanded this section of the manuscript to better explain the pipeline developed for calling sequence-based off-target discovery. We've now answered the reviewer's question regarding the grouping of mismatches into \leq bp categories, to enable filter selection, which is designed to ensure that there is overlap between the categories. We believe that this is now made clear in the manuscript.

Detection of Cas9 off-target cleavage should be discussed in greater depth in the text. How does the number of mismatches in off-target sites affect break level at these sites? What is the impact of the position of these mismatches related to the seed sequence? How does this compare to existing literature related to cleavage at off-target sites?

We have now expanded our reporting and discussion of this important aspect of the manuscript.

There are no references to Supplementary Figure 6b and 6c in the text.

This section has now been restructured and all Figures and Supplementary Figures are now referred to appropriately.

What was the rationale for using filter condition 7 for subsequent analysis shown in Figure 3a?

This has now been explained in terms of the optimal selection of maximum number of off-targets reported with lowest false discovery rate (See Supplementary Figure 6A).

Based on Supplementary Figure 7, more off-target sites were detected using BLISS than any other method shown here, including INDUCE-seq. Why then do the authors state that INDUCE-seq 'significantly outperforms' BLISS?

We believe that the reviewer has misread or misinterpreted the Venn diagrams shown in this figure. The highest number of 'off-targets' is reported by the *in vitro* method Circle-seq (labelled outside the Venn). We believe that this addresses the reviewer's question. However, we have also improved the text to clear up any discrepancy.

How many libraries were pooled for a single sequencing run? The authors claim that INDUCE-seq offers improved cost effectiveness, but show no direct comparisons of cost between different DSB mapping techniques. We have now explained more clearly why INDUCE-seq is more cost effective than other methods. The primary reason is simply the significant amount of superfluous sequencing required following PCR amplification of labelled breaks. Every read obtained by INDUCE-seq is derived from a single break present in the sample. The sequencing cost alone is therefore dramatically reduced. There are other economies related to the initial number of cells required in the initial cell sample. We don't conduct a side-by-side comparison of cost with other methods, but Figure 3a and b alone illustrates our point convincingly. These aspects are now addressed in the manuscript.

Line 428: 'High Capacity flow cell' should presumably read 'High Output flow cell'?
This has now been corrected in the Methods.

Please indicate the length of the index present in the P7 adapter.
We have now corrected this in the Methods

If DNA ends have a 3' overhang rather than 5' overhang, how is the fill in reaction going to work? Are the ends blunted in this case?
Yes, the ends are blunted as described in the manuscript.

Can the authors explain better why they use the half-functional P7 adapter in their method?
We have revised the explanation of this in the text and we believe that this is now clear.

Reviewer #2 (Remarks to the Author):

The Dobbs et al. manuscript proposes INDUCE-Seq, a novel method for mapping DSBs by sequencing. INDUCE-Seq does not include a typical enrichment and amplification step, instead relying on the cluster generation procedure, an integral step of Illumina sequence, to achieve also selection of reads originating from DSBs. This is an elegant idea, which allows for streamlining of DSB mapping via sequencing. We are pleased that the reviewer acknowledges the advantages of eliminating PCR amplification of labelled breaks to simplify and improve the accuracy of DSB mapping by sequencing.

An important weakness of the paper though are the exaggerated claims of significance and novelty and the fact that some of the conclusions are unsupported by the data. We acknowledge the shortcomings of the original manuscript, and trust that the revisions we have made now justify our claims. Many of the points raised by Reviewer 2 are the same concerns as those raised by Reviewer 1, but expressed in an alternative way. Therefore, we hope that Reviewer 2 will also read our response to reviewer 1 for additional relevant information.

For example, the authors incorrectly claim that precise DSB quantification (reported in Zhu et al., Nature Communications, 2019) or detection of low-frequency, endogenous DSBs (reported in Biernacka et al., Communications Biology 2018) were not previously possible. More generally, the authors both describe and visualize (Figure 1 bottom right) existing DSB mapping methods as noisy, which was true several years ago (e.g. original BLESS protocol, 2013), but is not true currently. Recent DSB mapping methods exhibit low noise (e.g. END-Seq allows to detect breaks after 1:10,000 dilution (Canela et al., 2016), i-BLESS after 1:100,000 (Biernacka et al., 2018)); they were also proven to be able to detect even very rare endogenous breaks, e.g. breaks induced by G-quadruplexes in vivo (i-BLESS). Precise DSB quantification is also now achievable through a straightforward approach, described in Nature Communications (Zhu et al., 2019). In our initial report, we focused our attention on the advantages of eliminating the PCR amplification of labelled DSB in cells and the significant noise associated with, and introduced by, this process to the break sequencing output. It is the case that the methods listed above by the reviewer have all contributed in different ways to improving the measurement of DSBs by correcting for, or normalising the breaks measured by other PCR-dependent sequencing methods. However, all these methods are still dependent on a PCR amplification process that breaks the relationship between true breaks labelled in cells and the sequencing reads derived from them following amplification (we also refer the reviewer to our response to Reviewer1). Furthermore, there are certain limitations of the methods mentioned above that are circumvented by INDUCE-seq. These issues have now been included in the revised manuscript and explained in further detail in the sections noted below. We trust that our inclusion of these additional sections are viewed positively, allowing us to demonstrate the significant advance that INDUCE-seq achieves, rather than being a criticism of the above methods, which attempt to correct for the various issues associated with PCR-based methods.

Moreover, eliminating DSB enrichment and PCR amplification from current DSB mapping procedures, although definitely an improvement, is not as impactful as the authors suggest, since it is not true that the amplification is causing high-noise level in the newer methods (END-Seq, i-BLESS).

Both the above-mentioned methods employ improvements to the cell/nuclei preparations and the DSB labelling strategy, which contributes to reducing the production and detection of non-physiological breaks in cells. However, they both still employ a downstream PCR amplification step that will inevitably break the relationship between true breaks present in the cell, and reads obtained following sequencing, which introduces noise. It is worth reiterating that INDUCE-seq is agnostic to the early stages of cell preparation and break labelling, meaning that these early stage break labelling protocols could be used in combination with INDUCE-seq (see also comments to Reviewer1).

Moreover, eliminating streptavidin DSB-enrichment and PCR amplification is not huge in terms of saved time or costs. Typical DSB mapping protocols are multiday (DSBcapture ~8 days, BLISS ~4 days, i-BLESS ~5 days), while streptavidin enrichment takes half an hour (3 hours with removal), PCR amplification - 2h, both are inexpensive.

The cost savings we refer to relate predominantly to the significant reduction in the actual number of sequence reads generated by the various different methods. On a cost-per-break basis, new Figure 3a and b demonstrates the stark contrast between sequencing multiple copies of amplified breaks versus the sequencing of 1 read per break as is the case for INDUCE-seq. We have modified the text extensively to better illustrate this point (see also response to Reivewer1).

The authors also repeatedly claim that their method is quantitative, that they accurately quantified breaks of various frequencies, etc – offering no validations whatsoever of these claims. It is not even indicated how the numbers of DSBs per cell plotted in Fig. 2c were computed. Estimating DSBs per cells by dividing reads originating from unique breaks by the number of cells used, as proposed in BLISS (my guess how the numbers in Fig. 2c were calculated) may lead to very inaccurate results, as shown in Zhu et al, Nat. Comm. 2019, Fig. 2g. This is an important point that requires careful explanation. We also refer the reviewer to our response to Reviewer1 on this same topic. The reviewer correctly states that we estimate DSBs per cell by dividing the total number of reads we obtained by the total number of cells used in the starting sample. This is indeed the way that BLISS also estimates this number, albeit after having to correct for PCR amplification by UMI correction, which is not a trivial undertaking. This simple calculation is uniquely valid when using INDUCE-seq data, due to the precise 1 read to 1 break relationship that is inherent to the INDUCE-seq method. Significantly, this is achieved without the need for any type of error correction. The reviewer notes that Zhu et al showed that this simple calculation using BLISS data does not correlate with the breaks per cell estimated by qDSB-seq. The reviewer appears to accept that the qDSB-seq breaks/cell estimate is likely to be correct because it appears to correlate well with the number of foci observed by gamma H2A-X break labelling also observed in DivA cells. However, there are important caveats to take note of when normalising for breaks using this method. qDSB-seq uses a restriction enzyme induced DSB spike-in to normalise and quantify the breaks present in the experimental

Zhu et al, 2019 Figure2e

samples. This method relies on engineered cell-lines, or the introduction of a specific type of exogenous DSB spike-in (eg NotI enzyme-induced breaks). The spike-in is considered representative of the full range of break types that are present in the experimental cell sample. This then permits the calculation of a correction factor that is subsequently used for normalisation of all break types present in the sample. An example of a linear relationship that can be observed for a spike-in of NotI-induced breaks is shown on the left and is taken from Figure 2e of the Zhu et al 2019 paper. However, the authors state that a linear relationship does not hold for both higher and lower-end cutting efficiencies observed at specific NotI restriction sites seen outside of the range of cutting efficiencies shown in Figure 2e of their report. Furthermore, Zhu et al also reports the loss of a proportional relationship between cutting efficiencies and labelled reads for a number of different enzymes with varying cutting efficiencies. This is clearly demonstrated in their Supplementary Figure 3 as shown below. This figure confirms our own observations now reported in the revision that the characteristics of the different types of breaks induced varies extensively throughout the genome. The corollary to this limits the application of this technique to normalising the break numbers between different experiments, but has limited application for use within a sample to estimate the numbers of breaks per cell. Whilst we feel it is worth mentioning this nuance here, the key point is that such error corrections are unnecessary when using INDUCE-seq

Zhu et al, 2019 Supplementary Figure 3. Employment of extremely low or high cutting efficiencies results in loss of proportional relationship between the labeled reads and cutting efficiencies at enzyme cutting sites. Pearson correlation (R) between cutting efficiencies and the labeled reads was calculated for all sites recognized by a given enzyme. From left to right, AsiSI, BamHI, and NotI-treated samples are shown.

The authors seem to believe that they achieved the perfect 100% efficiency of “1DSB=1 sequenced read”, but they did not attempt to verify it in any way.

This is a misconception. What we demonstrate is that 1 sequencing read is equivalent to 1DSB labelled in the cell. That is true by the design of the method. We acknowledge, however, that the opposite is not always true, meaning that we are not claiming the 100% efficiency as stated by the reviewer. This is an important nuance, which we have now addressed in the revision.

Detecting breaks during sequencing is a complex, multi-stage process with many challenges. First, DSB ends have to be labeled efficiently. It is not clear how efficiently INDUCE-Seq is doing it. For example, it has been shown that digestion with proteinase K before labeling (not used in INDUCE-Seq) improves the efficiency of labeling substantially (Biernacka et al., 2018, i-BLESS). Second, introduction of artefactual breaks has to be minimized. Initially, it was believed that fixation with formaldehyde helps to protect DNA integrity, but one of the innovations of the END-Seq (Canela et al., 2016) was to eliminate formaldehyde fixation, which allowed to significantly reduce noise. Usage of formaldehyde was again linked with higher noise in Biernacka et al., 2018. In INDUCE-Seq, fixation with 4% formaldehyde is used.

As previously mentioned, all these variations and improvements to break-labelling methods do not affect the downstream amplification subsequently employed. Furthermore, these labelling strategies can equally be applied to the INDUCE-seq protocol.

Thirdly, not every DNA fragment will lead to a correct cluster on an Illumina flowcell. Sometimes more than one read will be present in a cluster (and such clusters are likely to be later rejected), sometimes one read will result in two neighboring clusters, etc. The authors allude to these kinds of problems only once and rather cryptically in the methods section (“optical duplicates were removed”), but they do not quantify how efficient really their labeling procedure is, nor do they mention it anywhere in discussion as a potential limitation.

The points raised here also apply to all NGS methods and we have shown that INDUCE-seq libraries perform as expected. The optical duplicate removal procedure is fully explained in the Methods section. As described in the revised manuscript the labelling procedure used is based on a modified version of the BLISS protocol and performs similarly. However, again, INDUCE-seq is agnostic to break labelling methods and can variations on the labelling protocol can be applied.

Moreover, the comparison with DSBcapture shown in Fig. 2c and 2d is meaningless and even misleading. It is true that percentage of reads mapped to restriction site of an endonuclease when fully digested is used to compare efficiencies of DSB detecting methods, but it only makes sense when the same enzyme is used for both methods! As the authors state correctly, observed DSBs not mapping to restriction sites can include both endogenous DSBs and noise. Of course, depending on restriction enzyme, the same 10 endogenous DSBs will be a different percentage of a total signal, for restriction enzyme inducing likely 1,000s of DSBs and for an infrequent cutter. Here, since EcoRV has substantially fewer restriction sites it is unsurprising that percentage of reads mapping to them are a bit lower than for HindIII; the authors will likely get the same conclusion if they used both HindIII and EcoRV themselves. The figure 2c and d had to be removed as meaningless, or INDUCE-Seq has to be repeated with EcoRV, which is an easily commercially obtainable restriction enzyme.

We respectfully disagree with this assertion. We consider it reasonable to compare two different restriction enzymes in this way. However, we have modified the text referring to this section to better communicate the comparisons made. The key point is that INDUCE-seq performs similarly for detecting enzyme-induced breaks as DSBcapture, but using vastly reduced numbers of cells with greatly reduced sequencing costs.

Overall, removing PCR amplification from DSB mapping pipeline, as the authors propose, is a good idea, since Illumina technology allows it. It is also likely that INDUCE-Seq is highly sensitive, as shown by results of amplicon sequencing. However, it was not proven that it is superior to the current method, other than requiring less starting material. To be published, numerous inaccuracies have to be corrected, some specific examples are listed below.

We agree with these points and have now extensively revised the manuscript accordingly. The key sections highlighted and listed below have now been addressed.

Lines 76-78 “Existing methodologies typically measure DSBs reliably only when they exist at recurrent ‘hotspots’, or when they are induced at defined genomic-locations by sequence-directed nucleases.” Not true, for examples see i-BLESS (2018), proof of reproducibility in qDSB-Seq (2019).

We don't make this claim in this way any longer, but we have revised the MS to better explain the key advances and to address certain limitations related to correcting for PCR amplification as described previously.

Lines 87-88 “This well-known phenomena makes the quantification of genomic DSBs impossible”. This statement is based on papers published in 2011-2015 and directly contradicted by newer results (qDSB-Seq, 2019).

We now include a description of qDSB and have modified our claims accordingly (also, see above).

Bottom right of Figure 1 is misleading, the authors suggest that the current DSB mapping methods are so noisy that based on the read mapping, one cannot even confidently tell where the DSB was located, which is not true for well-defined DSBs tested with the newer methods (e.g. Fig 1d of i-BLESS).

We have acknowledged that there are modified labelling methods that can reduce the noise in the system, which can result in higher levels of sensitivity when analysing the break data at known, well-defined restriction sites (eg iBLESS). However, INDUCE-seq makes it possible to sensitively detect breaks in their accurate proportions, with no *a priori* knowledge of their position in the genome.

Lines 122-125 “Therefore, PCR error-correction, such as a unique molecular identifier (UMI) is required to attempt DSB quantification”. DSB quantification can be simply achieved by employing spike-in DSB used to calibrate studied breaks, as validated in Zhu et al. Nat. Comm. 2019. That approach proved much more accurate than an earlier attempt to use UMIs (BLISS) (Zhu et al. 2019).

We have modified this section, to acknowledge the improvements made by these methods, but have also highlighted some of the limitations of these approaches and explained how INDUCE-seq circumvents them.